# Nuclear Localization of Human SOD1 in Motor Neurons in Mouse Model and Patient Amyotrophic Lateral Sclerosis: Possible Links to Cholinergic Phenotype, NADPH Oxidase, Oxidative Stress, and DNA Damage

**DOI:** 10.3390/ijms25169106

**Published:** 2024-08-22

**Authors:** Lee J. Martin, Shannon J. Koh, Antionette Price, Dongseok Park, Byung Woo Kim

**Affiliations:** 1Department of Pathology, Division of Neuropathology, Johns Hopkins University School of Medicine, 558 Ross Building, 720 Rutland Avenue, Baltimore, MD 21205-2196, USAdpark58@jhmi.edu (D.P.);; 2Pathobiology Graduate Program, Johns Hopkins University School of Medicine, 558 Ross Building, 720 Rutland Avenue, Baltimore, MD 21205-2196, USA; 3Department of Neuroscience, Johns Hopkins University School of Medicine, 558 Ross Building, 720 Rutland Avenue, Baltimore, MD 21205-2196, USA; 4Department of Anesthesiology & Critical Care Medicine, Johns Hopkins University School of Medicine, 558 Ross Building, 720 Rutland Avenue, Baltimore, MD 21205-2196, USA; 5Texas Health Presbyterian Hospital, Dallas, TX 75231, USA

**Keywords:** motor neuron, peroxynitrite, comet assay, genome integrity, chromatin

## Abstract

Amyotrophic lateral sclerosis (ALS) is a fatal disease that causes degeneration of motor neurons (MNs) and paralysis. ALS can be caused by mutations in the gene that encodes copper/zinc superoxide dismutase (SOD1). SOD1 is known mostly as a cytosolic antioxidant protein, but SOD1 is also in the nucleus of non-transgenic (tg) and human SOD1 (hSOD1) tg mouse MNs. SOD1’s nuclear presence in different cell types and subnuclear compartmentations are unknown, as are the nuclear functions of SOD1. We examined hSOD1 nuclear localization and DNA damage in tg mice expressing mutated and wildtype variants of hSOD1 (hSOD1-G93A and hSOD1-wildtype). We also studied ALS patient-derived induced pluripotent stem (iPS) cells to determine the nuclear presence of SOD1 in undifferentiated and differentiated MNs. In hSOD1-G93A and hSOD1-wildtype tg mice, choline acetyltransferase (ChAT)-positive MNs had nuclear hSOD1, but while hSOD1-wildtype mouse MNs also had nuclear ChAT, hSOD1-G93A mouse MNs showed symptom-related loss of nuclear ChAT. The interneurons had preserved parvalbumin nuclear positivity in hSOD1-G93A mice. hSOD1-G93A was seen less commonly in spinal cord astrocytes and, notably, oligodendrocytes, but as the disease emerged, the oligodendrocytes had increased mutant hSOD1 nuclear presence. Brain and spinal cord subcellular fractionation identified mutant hSOD1 in soluble nuclear extracts of the brain and spinal cord, but mutant hSOD1 was concentrated in the chromatin nuclear extract only in the spinal cord. Nuclear extracts from mutant hSOD1 tg mouse spinal cords had altered protein nitration, footprinting peroxynitrite presence, and the intact nuclear extracts had strongly increased superoxide production as well as the active NADPH oxidase marker, p47phox. The comet assay showed that MNs from hSOD1-G93A mice progressively (6–14 weeks of age) accumulated DNA single-strand breaks. Ablation of the *NCF1* gene, encoding p47phox, and pharmacological inhibition of NADPH oxidase with systemic treatment of apocynin (10 mg/kg, ip) extended the mean lifespan of hSOD1-G93A mice by about 25% and mitigated genomic DNA damage progression. In human postmortem CNS, SOD1 was found in the nucleus of neurons and glia; nuclear SOD1 was increased in degenerating neurons in ALS cases and formed inclusions. Human iPS cells had nuclear SOD1 during directed differentiation to MNs, but mutant SOD1-expressing cells failed to establish wildtype MN nuclear SOD1 levels. We conclude that SOD1 has a prominent nuclear presence in the central nervous system, perhaps adopting aberrant contexts to participate in ALS pathobiology.

## 1. Introduction

ALS is a fatal adult-onset neurodegenerative disease that kills patients generally within 3 to 5 years after symptoms begin [1,2]. The disease is characterized by progressive weakness, muscle atrophy, spasticity, paralysis, and respiratory arrest [1,2]. These symptoms are related to the degeneration of motor neurons (MNs) in the cerebral cortex, brainstem, and spinal cord [1,3]. ALS occurs mostly sporadically, with unclear etiology, but the disease can be inherited within families [2]. Familial forms of ALS (fALS), ~10% of all cases, have autosomal dominant or recessive inheritance of heterogeneous gene mutations [2,3]. Approximately 20% of fALS cases are linked to mutations in the *superoxide dismutase 1* (*SOD1*) gene [4,5], which encodes a metalloenzyme that converts the highly toxic superoxide anion to molecular oxygen and hydrogen peroxide [6]. These mutations might confer a toxic gain of function to the protein, including aberrant oxidative chemistry catalysis and protein binding, rather than a loss of dismutase activity [4,7,8]. Wildtype SOD1 has been implicated in causing sporadic forms of ALS through altered function driven by posttranslational modification [9,10,11,12].

Yet neither the underlying causes of MN degeneration nor their extraordinary vulnerability to the disease are understood in human ALS and animal models of ALS, despite decades of intensive research [2,13,14,15,16,17,18]. Abnormal protein aggregation and cytoplasmic inclusion formation, excitotoxicity, mitochondrial dysfunction, axon abnormalities, target deprivation with axonal dying-back, and oxidative stress from reactive oxygen species or reactive nitrogen species have been implicated in causing MN degeneration in ALS [15,16,17]. With human SOD1 (hSOD1)-related fALS and sporadic ALS, controversy surrounds whether mutant and wildtype proteins acquire a toxic property or function through misfolding and aggregation, mislocalization, or aberrant oxidative chemistry [18]. Nitric oxide (NO) and peroxynitrite have been implicated in the process of MN death in cell culture models [19,20] as well as in the process of MN degeneration in vivo in transgenic (tg) mice expressing hSOD1-G93A and in non-tg rodent axotomy models [21,22,23,24]. Mitochondria garner much attention in studies of ALS pathobiology [15,22,25,26,27,28], but interest in understanding the mechanisms involving the MN nucleus is growing [18,29,30,31].

Nuclear pathology occurs in ALS. Abnormal nuclear inclusions have been seen to be positive for ubiquitin, promyelocytic leukemia protein, and ataxin-3 in human ALS MNs [32]. Mutations in the *fused sarcoma* (*FUS*) gene are associated with some fALS cases. FUS protein functions in DNA repair and undergoes nucleocytoplasmic shuttling, but some FUS mutants have impaired nuclear localization, and they accumulate in the cytoplasm [33,34]. Human sporadic ALS cases have elevated levels of 8-hydroxy-2-deoxyguanosine (8OHdG)—a robust marker of oxidatively damaged DNA [35]—in vulnerable CNS regions, as detected biochemically [36]. We found 8OHdG in human ALS cases directly in the upper and lower MNs with activated p53 [37]. Human ALS MNs also accumulate single-stranded DNA and DNA double-strand breaks [14,38]. The protein levels and catalytic activity of the DNA base excision repair enzyme apurinic-apyrimidinic endonuclease-1 (APE1) appeared to increase in human ALS-vulnerable CNS regions, such as the primary motor cortex (precentral gyrus) and ventral horn of the spinal cord [39], but, in the frontal cortex, APE1 appeared decreased [40]. hSOD1-G93A tg mice also accumulate 8OHdG in the spinal cord [41], and MNs have increased DNA single- and double-strand breaks [22].

Sau and colleagues [42] elegantly described a possible role of nuclear SOD1 in ALS pathogenesis, further supporting DNA damage as an upstream mechanism [43,44,45,46,47]. Subsequently, we identified that nuclear hSOD1 presence in tg mice with mutant variants is related to the aberrant subnuclear compartmentation of survival motor neuron-1 (SMN1)—also known as gemin 1—complexes that function in RNA processing [48]. However, the key DNA repair enzyme 8OHdG glycosylase, an initial critical enzyme involved in base excision repair [46], remained in the nucleus of MNs [48]. A study of peripheral blood mononuclear cells of sporadic ALS cases then suggested that phosphorylated SOD1 in the nucleus is important for protection against oxidative DNA damage [49]. However, nuclearopathy and DNA damage, as mechanisms in ALS, need further support as disease drivers. For example, it is not known whether SOD1 has a nuclear presence in glial cells, in addition to MNs, and if nuclear SOD1 is mislocalized in disease and associates with other potential disease mechanisms, such as DNA damage and oxidative stress related to superoxide and NADPH oxidase. Such perturbations might help to explain the selective vulnerability of MNs in ALS [1,2,3]. In addition, SOD1 has been described in the nucleus of mostly mouse experimental animal and cell systems, but scant work on the nuclear localization of SOD1 has been conducted on human cells and ALS-relevant systems [50], notably neurons.

This study provides new information on the cellular localization of nuclear SOD1 in hSOD1 tg mice and the novel potential mechanisms of experimental ALS involving the MN nucleus, NADPH oxidase, and DNA damage accumulation. The translational relevance of this work was evidenced by the SOD1 nuclear presence in neurons and glia of the human brain as well as elevated neuronal nuclear SOD1 and inclusion formation in ALS cases. In human-induced pluripotent stem (iPS) cells [38,51], we show the nuclear presence of SOD1 during cellular-directed-differentiation to MNs with fALS-linked SOD1 mutations and found that mutant variants of SOD1 behave differently than wildtype SOD1. Overall, nuclear SOD1 appears involved in ALS pathobiology, but it is unlikely to be the sole player in the complex milieu of the nucleus.

## 2. Results

### 2.1. hSOD1 Nuclear Localization in Tg Mouse Spinal Cord Is Cell-Type Differential and Disease-Related

Human-specific SOD1 antibody [52,53] was used to localize SOD1 in tg mice expressing the hSOD1-WT and -G93A variants. Age-matched mice were evaluated at 6, 8, 10, and 14 weeks of age. Non-tg mice were not included in this experiment because they do not express hSOD1 [22]. hSOD1 was localized in specific cell types of the spinal cord using cell type-specific markers. In hSOD1-WT mice, MNs, identified by choline acetyltransferase (ChAT), generally had stable nuclear positivity from 6–10 weeks; then, a small fraction of MNs had a loss of positivity at 14 weeks of age (Figure 1A–C and Figure 2A). The hSOD1-G93A mice, in contrast, showed a progressive age-related decline in the number of MNs with nuclear hSOD1 positivity despite preservation of the cytoplasmic ChAT phenotype, though it was severely vacuolated (Figure 1D–F and Figure 2A). A subset of spinal cord interneurons was visualized by parvalbumin [22], which is enriched in the cytoplasm and nucleus (Figure 3) [22]. While these neurons had nuclear hSOD1 in hSOD1-WT and -G93A mice (Figure 3), parvalbumin positivity in the nucleus of these neurons persisted throughout the course of the disease in the hSOD1-G93A mice (Figure 2B).

Spinal cord astrocytes and oligodendrocytes were visualized with their respective markers, glutamine synthetase (GS, Figure 4) and 2′,3′-cyclic nucleotide 3′-phosphodiesterase (CNPase, Figure 5), in the hSOD1-WT and -G93A mice. GS and CNPase were used because these markers are useful for clearly identifying astrocyte and oligodendrocyte cell bodies, respectively, without too much of the associated neuropil immunoreactivity that can confound cell counting. Non-tg mice were not used because their cells would not be hSOD1-positive. In the hSOD1-WT mice at 6 weeks of age, ~60% of the astrocytes had hSOD1 nuclear positivity (Figure 4A–D); this did not change through 14 weeks of age (Figure 6A). In contrast, the hSOD1-G93A mice at 6 weeks of age, compared to the age-matched hSOD1-WT mice, had a significantly lower number (~30%, *p* = 0.009) of astrocytes with nuclear hSOD1 positivity (Figure 4E–H), and by 10 weeks of age, a non-significant trend upward (*p* = 0.05, compared to hSOD1-G93A mice at 6 weeks) was found in the number of astrocytes with nuclear hSOD1 (Figure 6A). Oligodendrocytes had different overall profiles compared to the astrocytes (Figure 5). In the hSOD1-WT mice at 6 weeks of age, ~80% of the oligodendrocytes had hSOD1 nuclear positivity (Figure 5A–D); this percentage did not change through 14 weeks of age (Figure 6B). In stark contrast, the hSOD1-G93A mice at 6 weeks of age had only ~12% of oligodendrocytes with nuclear SOD1 (Figure 5E–H and Figure 6B). This was a highly significant difference (*p* < 0.0001) compared to the age-matched hSOD1-WT mice (Figure 6B). The percentage of spinal cord oligodendrocytes with nuclear hSOD1 increased significantly with disease progression (Figure 6B). The hSOD1-G93A mice at 10 and 14 weeks showed ~45–50% (Figure 6B, *p* = 0.008) of oligodendrocytes with nuclear hSOD1 compared to 6 weeks of age.

### 2.2. Subcellular Fractionation of Brain and Spinal Cord of hSOD1-G93A Mice Shows Differential Enrichment of hSOD1 in Nuclear Chromatin Extracts

We used rigorously characterized subcellular fractionation by differential detergent extraction to study the compartmentalization of hSOD1 (Figure 7) in the cytoplasmic extract (CE), soluble nuclear extract (sNE), and chromatin nuclear extract (cNE). The CE was enriched in glyceraldehyde phosphate dehydrogenase (GAPDH) (Figure 7). The cNE was enriched in methyl-CpG-binding protein 2 (MeCP2) and histone H3 (Figure 7). Histone H3 was highly specific for the cNE (Figure 7), consistent with the presence of chromatin only in this fraction. In the brains (Figure 7A) and spinal cords (Figure 7B) of hSOD1-G93A tg mice, hSOD1 was most enriched in the CE. The Non-tg control mice had no hSOD1 immunoreactivity (Figure 7A,B), thus validating the human-specific SOD1 antibody because it does not detect endogenous mouse SOD1. hSOD1 protein was detected in the sNE at about 25% of the CE level in both the brain and spinal cord. In the cNE of the brain, there was only a faint trace of hSOD1 (Figure 7A). In contrast, in the cNE of the spinal cord, hSOD1 was detected at a level of about 25% of the CE (Figure 7B).

### 2.3. Isolation of an Intact Nuclear Fraction of Spinal Cord for Acute Live-Nucleus Analyses of Oxidative Stress and DNA Damage

We used a protocol to isolate intact MNs (see Appendix A) and their nuclei from the ventral horn of the mouse spinal cord using microdissection [54,55], proteinase K digestion, and sucrose gradient centrifugation (Figure 8A) [48]. The nucleus-enriched fraction was verified by staining for DNA (DAPI), nuclear membrane protein (LAP2), and nuclear matrix protein (NeuN) (Figure 8B). Spinal cord ventral horn MN-enriched nuclei were isolated, lysed, and subjected to SDS-PAGE and Western blotting for nitrotyrosine (Figure 8C) for the detection of protein nitration that is a peroxynitrite footprint [56]. hSOD1-G93A tg mouse ventral horn spinal cord neuronal nuclei (compared to control mouse spinal cord) showed increased nitration of bands of proteins at ~200 kDa and ~15–20 kDa, but lower nitration of proteins at ~30–32 kDa (Figure 8C). We then used hydroethidium dye [55,57] to detect superoxide in living, intact MN nuclei (Figure 8D–G). In hSOD1-WT mice, MN nuclei had modest but detectable levels of superoxide at 6–10 weeks of age; between 10 and 14 weeks, a significant increase (*p* = 0.01) in superoxide production was seen (Figure 8H). However, hSOD1-G93A mouse MN nuclei, in contrast, had a progressive age-related significantly increased nuclear superoxide production compared to the hSOD1-WT mice (Figure 8H). Notably, the hSOD1-G93A mice at 10 weeks of age had higher levels of superoxide production (Figure 8H) compared to the age-matched hSOD1-WT mice (*p* = 0.002) and hSOD1-G93A mice at 6 weeks of age (*p* = 0.007).

### 2.4. MNs from Spinal Cord of Tg Mice Have hSOD1 Variant-Related Accumulation of DNA Single-Strand Breaks

Because we detected peroxynitrite and superoxide signatures in the MN nuclei of hSOD1 tg mice, we assayed for DNA damage. We used the comet assay (Figure 9) because we have validated this approach for MNs [54,55]. In this single-cell gel electrophoresis assay, cells or nuclei are embedded in agarose gels and subjected to pH 13 electrophoresis to elute the DNA with single-strand breaks, seen as comets after ethidium bromide staining (Figure 9B–F). Nuclei with intact genomic DNA have the ethidium staining concentrated in a round nucleoid without tails (Figure 9A). Nuclei with low DNA strand breaks have emergent tails from the eluted DNA for the nucleoid (Figure 9B), and those nuclei with greater amounts of DNA single-strand breaks have more obvious tails (Figure 9C). Non-tg mice at 6, 10, and 14 weeks of age had few MN comets (Figure 9D,G). MNs from the hSOD1-WT mice at 14 weeks of age had significantly (*p* < 0.001) more DNA damage than the age-matched non-tg mice and had significantly (*p* = 0.03) more comets compared to the 6-week-old hSOD1-WT mice (Figure 9E,G). MNs from the hSOD1-G93A mice at 6 weeks had significantly (*p* = 0.002) more DNA damage than the age-matched hSOD1-WT mice (Figure 9E,G) and had significant (*p* = 0.02) age-related accumulation of DNA damage within the hSOD1-G93A mouse groups (Figure 9G).

### 2.5. NADPH Oxidase Has a Role in Disease in hSOD1-G93A Mice

Our data on protein nitration (Figure 8C), superoxide production (Figure 8D–H), and DNA single-strand break accumulation (Figure 9) suggest that oxidative DNA damage could be involved in the disease mechanisms in ALS mice. Mitochondrial mechanisms of oxidative stress in ALS have been explored exhaustively [17,58]. NADPH oxidase is another prominent source of superoxide and is localized in the nucleus, as well as other more often studied cellular locations [59,60]. The role of NADPH oxidase in ALS is controversial, but compared to mitochondria, little work has been conducted on NADPH oxidase, particularly the potential nuclear mechanisms, in ALS. Regarding the role of NADPH oxidase in the hSOD1-G93A mouse model of ALS, there is both work that supports involvement [57] as well as data that does not support involvement [61].

We used Western blotting to measure p47phox and Ras-related C3 botulinum toxin substrate-1 (RAC1), both critical components of NADPH oxidase [59], in spinal cord ventral horn MN-enriched nuclear fractions of non-tg and hSOD1-G93A mice at 6, 8, 10, and 14 weeks of age (Figure 10A,B). For the hSOD1-G93A mice, the 6- and 8-week ages offered presymptomatic coverage of the disease, and the 10 and 14 weeks, covered symptomatic disease. RAC1 levels remained invariant in all groups (Figure 10A). In non-tg mice, the p47phox levels were stable in mice from 6 to 14 weeks of age (Figure 10B). In contrast, the hSOD1-G93A mice had highly significant elevations in their p47phox levels compared to their age-matched non-tg counterparts (Figure 10B).

We directly interrogated NADPH oxidase as a mechanism of disease in mouse ALS models by conducting genetic and pharmacological experiments. hSOD1-G93A mice with complete elimination of p47phox (homozygous deletion of the *NCF1* gene) lived, on average, significantly (*p* < 0.00001) longer (~25%) than the littermate hSOD1-G93A mice with p47phox (Figure 10C). Moreover, when hSOD1-G93A mice, with their p47phox genes intact, were treated with apocynin (10 mg/kg (ip) daily starting at 7 weeks of age), they also had an ~25% significantly increased (*p* < 0.0001) mean lifespan compared to the vehicle-treated littermate hSOD1-G93A mice (Figure 10D). The hSOD1-G93A/p47phox^−/−^ mice had significantly less (*p* = 0.0002) MNs with DNA single-strand break accumulation compared to the tg mice with p47phox at 10 weeks of age (Figure 10E). Similarly, 10-week-old hSOD1-G93A mice treated with apocynin had significantly fewer (*p* = 0.002) MNs with DNA damage accumulation compared to the vehicle-treated tg mice (Figure 10E).

### 2.6. SOD1 Localizes the Nucleus in MNs and Glia in Human CNS: Nuclear Presence Increases in ALS Upper and Lower MNs

We used immunoperoxidase immunohistochemistry to localize SOD1 in postmortem human ALS and age-matched control motor cortex and spinal cord (Table 1). Human neurons and glia in the control and ALS cases showed SOD1 nuclear positivity (Figure 11). In the control group’s motor cortex, layer 5 Betz cells—typified by their large size and pyramidal shape—had cytoplasmic SOD1 immunoreactivity interspersed with prominent Nissl substance but had scant nuclear SOD1 immunoreactivity, appearing inconspicuously as a few granules (Figure 11A,G). The glial cells also had nuclear SOD1 immunoreactivity in the motor cortex of the control humans (Figure 11A,H). The control human spinal cord MNs were large, grand, and robust (some 50–60 µm in large axis diameter) with cytoplasmic SOD1 immunoreactivity and low nuclear SOD1 immunoreactivity (Figure 11C). In the ALS motor cortex, layer 5 Betz cells were generally attritional and consistently had prominent nuclear SOD1 immunoreactivity (Figure 11B,G). Many layer 5 neurons in the ALS cases had SOD1 nuclear inclusions (Figure 11E). Glial cells in the motor cortex of ALS humans, though sometimes positive for nuclear SOD1 immunoreactivity, were often also attritional (Figure 11B,H). The remaining lateral cell group spinal MNs in the lumbar and cervical enlargements in the ALS cases were smaller than the control MNs and had conspicuous SOD1 nuclear immunoreactivity in a pattern that was reticulated and chromatin-like (Figure 11D). Many spinal MNs in the ALS cases had SOD1-positive nuclear inclusions (Figure 11F). Some glia were positive for SOD1, and many of these cells appeared as apoptotic profiles (Figure 11D).

### 2.7. Nuclear SOD1 Is Present in Human iPS Cells at Different Stages of Directed Differentiation to MNs, but Familial Mutant SOD1 ALS MNs Have Aberrant Subcellular Compartmentation of SOD1

SOD1 was localized in human iPS cells at different stages of directed differentiation to MNs (Figure 12). The cell markers verified the different stages of differentiation from a pluripotent stem cell (PSC, Figure 12A), neuroepithelial cell (NEP, Figure 12B), and motor neuroprogenitor (MNP, Figure 12C) to mature MNs (Figure 12D), which were identified with OCT3/4, SOX1, OLIG2, and ChAT, respectively (Figure 12). Subcellular fractionation and Western blotting of PSCs, NEPs, and MNPs showed similarities in the proportions of SOD1 in the cytoplasmic and nuclear extracts from SOD1-WT, SOD1-G93A, and SOD1-A4V cells (Figure 12E–G). In contrast, mature MNs with the SOD1-WT genotype differed dramatically from the SOD1-G93A and SOD1-A4V MNs in the proportions of SOD1 in the cytoplasmic and nuclear extracts (Figure 12E–G). For differentiated SOD1-G93A human MNs, SOD1 appeared depleted in the nuclear extract (Figure 12F). For differentiated SOD1-A4V human MNs, SOD1 also appeared depleted in the nuclear extract (Figure 12G), but overall, the SOD1 level was low in both the cytoplasmic and nuclear extracts compared to the SOD1-WT and SOD1-G93A MNs (Figure 12E–G).

## 3. Discussion

We used several different experimental model systems and the human brain and spinal cord to examine nuclear SOD1 in the context of ALS. In the mouse spinal cord, hSOD1 (mutant and WT variants) was present in ChAT-positive MN nuclei. The hSOD1-G93A mice MNs showed symptom-related loss of nuclear ChAT. hSOD1-G93A was less commonly found in spinal cord astrocytes and oligodendrocytes, but with disease progression, nuclei of the oligodendrocytes had increased mutant hSOD1 presence. Subcellular fractionation of the mouse brain and spinal cord identified mutant hSOD1 in soluble nuclear extracts. However, mutant hSOD1 was concentrated in the chromatin nuclear extract only in the spinal cord. Spinal cord ventral horn nuclear extracts from mutant hSOD1 tg mice presymptomatically had aberrant protein nitration, increased superoxide production, and an increased protein level of NADPH oxidase organizer p47phox, while MNs progressively accumulated DNA single-strand breaks. NADPH oxidase involvement in disease progression in the hSOD1-G93A mice was established by p47phox-encoding *NCF1* gene ablation and pharmacological inhibition with apocynin. SOD1 was also found in the nucleus of neurons and glia of the human brain and spinal cord; neuronal nuclear SOD1 was increased in the ALS cases, and SOD1 formed nuclear inclusions. Human iPS cells expressing SOD1-WT, -G93A, and -A4V had nuclear SOD1 during differentiation to MNs, but cells with mutant SOD1 failed to adopt wildtype MN nuclear SOD1 levels. We conclude that nuclear SOD1 in MNs is poised to participate in a newly delineated neuronal cell death pathway, engaged in the nucleus, and in the pathogenesis of ALS through a variety of possible mechanisms, including chromatin dynamics and DNA damage.

### 3.1. hSOD1-G93A MNs with Nuclear SOD1 Lose Their Cholinergic Phenotype of the Nucleus

While using ChAT, the enzyme that synthesizes acetylcholine that is used at the neuromuscular junction to engage excitation–contraction coupling [62], as a marker for spinal cord MNs in mice, we observed that immunoreactivity for ChAT was also present in the nucleus along with SOD1. Interestingly, the hSOD1-G93A mice showed a major disease progression-related loss of ChAT nuclear positivity, while the hSOD1-WT mice showed only a modest loss of nuclear ChAT during aging. Other nuclear proteins, such as DNA repair enzymes and snurportins, show stability in nuclear localization in hSOD1-G93A mouse MNs [48]; however, nuclear gemin1 (SMN1) does not, and nuclear gemins collapse [48]. Subsets of ventral horn spinal cord interneurons, identified by parvalbumin, appear to show preservation of their nuclear phenotypes (Figure 2B and Figure 3). Nuclear ChAT has been described before [63]. ChAT possesses a nuclear localization sequence and undergoes nucleocytoplasmic shuttling through the nuclear pore [64]. The function of nuclear ChAT is not clearly understood. ChAT might be a chromatin-binding protein that influences transcriptional regulation at baseline functioning and in response to cell stress [65]. ChAT nuclear downregulation or export is thought to be involved detrimentally in human aging and Alzheimer’s disease, where the magnocellular cholinergic neuron dysfunction in the basal forebrain is at risk of causing age- and disease-related memory and cognitive decline [64]. We found that hSOD1-G93A is a chromatin fraction-associated protein in the mouse spinal cord, so it is tempting to speculate that mutant hSOD1 could be displacing the normal association of ChAT with chromatin in MNs. Lower MNs, those innervating somatic and branchial arch-derived skeletal muscle, are the most vulnerable neurons in ALS [1,3], and they are also magnocellular cholinergic neurons. Additional work is needed to determine if the loss of the cholinergic nuclear phenotype in MNs is related to ALS pathogenesis.

### 3.2. Spinal Cord Oligodendrocytes in hSOD1-G93A Mice Adopt a Nuclear hSOD1 Phenotype during Disease

We discovered that a low percentage of oligodendrocytes in young hSOD1-G93A mice spinal cords had nuclear hSOD1 positivity, but as mice developed the disease, a strong hSOD1 nuclear phenotype emerged in the oligodendrocytes. This finding contrasted with the hSOD1-WT mice that had a relatively stable percentage (~70%) of oligodendrocytes with nuclear hSOD1 in the spinal cord ventral horn. The SOD1 antibody is human-specific (Figure 7) [48,54]; thus, endogenous mouse SOD1 detection is nil, so what is seen is WT or mutant variants of hSOD1. In the human postmortem ALS brain, we found nuclear SOD1 as exquisite inclusions in neuronal and glial nuclei. Some of these inclusions had an apparent chromatin association. We demonstrate a chromatin association of hSOD1 in the mouse spinal cord by subcellular fractionation, but the cell type cannot be deduced with these experiments. Previously, oligodendrocyte pathology in hSOD1-G93A mice was described using electron microscopy, though nuclear pathology was not mentioned [66]. Our findings accord with the elegant study of Forsberg et al. [67]. This group studied human postmortem cases of fALS with confirmed SOD1 mutations and sporadic ALS cases and found oligodendrocyte nuclear SOD1 using antibodies to misfolded protein [67]. Human brain postmortem SOD1 staining has limitations on its interpretation, given that these cases represent endstage human disease. Our finding in hSOD1-G93A mice, showing an age-related accumulation in nuclear hSOD1-positive oligodendrocytes, provides a system to vet the mechanistic role of nuclear SOD1 in cell death and in the pathogenesis of ALS in future studies.

### 3.3. Oxidative Stress within the Nucleus and DNA Damage Occur in MNs in hSOD1-G93A Mouse Spinal Cord

We used a protocol [48] to isolate the neuronal nuclei from the spinal cord ventral horn to study nuclear oxidative stress (Figure 8A,B). While the study of oxidative stress and its mechanisms, and the mechanisms involving DNA damage and proteinopathy, is plush in the context of aging and neurodegeneration in general [44,68,69], oxidative stress, specifically in the nucleus, is understudied. We found that some nuclear proteins show increased nitration in the hSOD1-G93A mouse spinal cord ventral horn MN-enriched fractions. Future mass spectroscopy experiments are necessary to identify these proteins. We also found nuclear fraction accumulation of NADPH oxidase subunit p47phox already at 6 weeks of age. hSOD1-G93A mice are presymptomatic at 6 weeks of age, symptomatic at about 10 weeks, and become completely paralyzed and die at ~16–18 weeks of age [70] with severe pathology to MNs [22,71,72]. Because NADPH oxidase is the main non-mitochondrial source of superoxide, we probed the isolated nuclei for superoxide production using hydroethidium [55,57]. We found significant increases in superoxide production in the hSOD1-G93A mouse MN nuclei early in the disease.

We examined the DNA damage in hSOD1-G93A mice for several reasons. First, we suspected a nuclear presence of peroxynitrite, which potently causes DNA strand breaks in MNs [54,55]. Second, we found elevations in nuclear p47phox. p47phox is part of the NADPH oxidase complex that generates superoxide. Third, we found that spinal cord MN nuclei generate higher levels of superoxide than the controls. Fourth, we have previously shown DNA single- and double-strand break accumulation in hSOD1-G93A mouse MNs using in situ histological methods [22]. In this study, we used an alternative approach, the comet assay, also known as single-cell gel electrophoresis [54]. Rather than being a tissue section-limited approach [22], the comet assay is a much larger scale cell population approach, using extracted MNs from entire spinal cord enlargements (in this case, the lumbar enlargements). We found that hSOD1-G93A mouse spinal cord MNs accumulate DNA single-strand breaks as early as 6 weeks of age. This timeframe parallels the nuclear presence of nitrated proteins, p47phox, and superoxide. An essential molecular component that we have not yet explored in this system is the nuclear presence of NO. NO is a rapidly diffusible uncharged free radical [73]. The cytotoxicity of NO is largely due to its reaction with superoxide to form peroxynitrite [73,74]. This reaction is catalyzed by SOD1 [75], which we find in the nucleus. Peroxynitrite is footprinted by protein nitration [75]. We found a peroxynitrite footprint in the nuclear fractions of hSOD1-G93A mouse MN nuclei early in the disease. Thus, a mutant SOD1 gain-of-function could be related to peroxynitrite formation within the nucleus, thusly identifying a possible new cell death pathway. Previously, we have shown that the knockout of constitutive and inducible forms of NO synthase can protect MNs from degeneration [22,76]. At present, we cannot place this interrupted NO-related neuroprotection directly in the context of the nucleus.

### 3.4. NADPH Oxidase Is Involved in the Disease Process in hSOD1-G93A Mice

We examined NADPH oxidase in our mouse model because this enzyme generates superoxide [59] and thus could be a key mechanism in SOD1-related ALS. We found that the genetic ablation of the p47phox-encoding *NCF1* gene in hSOD1-G93A mice extended the mean lifespan of hSOD1-G93A mice by about 25% and mitigated genomic DNA damage. Moreover, NADPH oxidase activity inhibition with systemic treatment of apocynin (10 mg/kg, ip) had similar positive effects. Thus, NADPH oxidase participates in the disease process in hSOD1-G93A mice because removing it allows the mice to live longer and attenuates genomic DNA damage accumulation. This shows that superoxide is at least partly related, perhaps through peroxynitrite formation, to the DNA single-strand break formation that is also related to the MN degeneration in these mice. This work provides evidence that DNA damage is causally related to MN degeneration rather than a secondary consequence of or downstream to degeneration in this mouse model of ALS. This concept is consistent with data showing that adenoviral-mediated enforced DNA repair can protect against the degeneration of MNs [77] and, with recent effective overviews [78] that embrace the narrative of DNA damage as an upstream mechanism of neurodegeneration in a variety of diseases.

Our results concerning NADPH oxidase in MN disease are mostly consistent with other, albeit limited, work. One study showed that NADPH oxidase components gp91phox and p67phox are upregulated in the hSOD1-G93A mouse total spinal cord; this was interpreted in the context of tissue inflammation [79] rather than in the context of MN autonomous mechanisms, as we do. We found an upregulation of p47phox, specifically in the MN nuclei from the lumbar spinal cord. Comparable to our results with nitrated proteins and superoxide production, Wu et al. [79] found increased protein oxidative damage in the total spinal cord. Moreover, while Wu et al. [79] found that genetic ablation of gp91phox (*NOX2* gene) increased the lifespan by about 10%, we found that genetic ablation of p47phox increased the mean lifespan by about 25%. Another study confirmed that the genetic deletion of gp91phox increased the lifespan of hSOD1-G93A mice and showed that the genetic deletion of p67phox (*NCF2* gene)—another subunit of NADPH oxidase [59]—increases the lifespan [80]. A follow-up study reported an increased superoxide production in the lumbar spinal cord tissue sections of hSOD1-G93A mice [57], while we isolated the MN nuclei from the lumbar spinal cord and showed increased superoxide production in the hSOD1-G93A mice. Harraz et al. [57] also showed that apocynin administration in the drinking water, starting at 14 days of age, increased the lifespan of hSOD1-G93A mice, while our apocynin administration was by intraperitoneal injection. Drinking water administration of apocynin is more relevant translationally. Interestingly, diapocynin (a covalent dimer of apocynin) treatment in drinking water, starting at 21 days of age, failed to extend the lifespans of the hSOD1-G93A mice [61]. The slight delay in the age of treatment initiation may be significant, but perhaps more importantly, diapocynin has much less CNS permeability than apocynin [61]. Overall, the earlier work and our new work identify NADPH oxidase as a molecular target in experimental neuronal cell death and human ALS and apocynin as a potential neurotherapeutic. However, the mechanistic understanding of apocynin’s therapeutic effect is challenging, as it appears not to be a direct catalytic inhibitor of NADPH oxidase, but instead, may inhibit the translocation of p47phox to gp91phox and p22phox that is needed for NADPH activation [59].

### 3.5. SOD1 Localizes to the Nucleus of MNs in Human ALS Brains and Spinal Cords and in Patient iPS Cell-Derived MNs in Cell Culture

We discovered that the upper and lower MNs in ALS cases accumulate SOD1 in the nucleus in vivo. In contrast, in glial cells, nuclear SOD1 was depleted, and some of these glial cells appeared apoptotic. Within the nucleus, SOD1 was localized to chromatin-like threads and apparent inclusions. Indeed, hSOD1 was found to be chromatin-associated in tg mice. We also demonstrate that SOD1 localizes to the nucleus in human iPS cells through all stages of differentiation from pluripotent stem cells to mature MNs. However, subcellular fractionation experiments showed that, at more immature stages of differentiation, there is SOD1 enrichment in the cytosolic compartment compared to the nucleus in wildtype cells, but, with cellular maturation, SOD1 presence in the soluble nuclear compartment accumulates progressively. This profile was not seen in the fALS iPS cells with G93A- and A4V-SOD1 mutations. Mutant SOD1 did not accumulate in the soluble nuclear compartment. It is possible that the mutant SOD1 is trapped in the insoluble or chromatin compartment in the iPS cell-derived MNs, and the nuclear SOD1 that accumulates in the nucleus of MNs in ALS tissue sections is aggregated and insoluble or chromatin-bound. Alternatively, the soluble mutant SOD1 could be exported rapidly from the nucleus. Nuclear phenotypes, including putative DNA damage with a surrogate marker, have been found in human ALS iPS cell-derived astrocytes [81]. Nuclear SOD1 could be participating in compensatory or etiopathologically significant processes. It could be compensatory from a redox standpoint or yet-to-be-discovered functions of SOD1 in the nucleus, such as chromatin modulation, transcription factor functioning, or the chromosome structure. Indeed, SOD1 has been described as a redox-sensitive transcription factor in yeast [82]. It could be pathological if it is collaborating with NADPH oxidase in the process of DNA damage or chromosome disruption or if SOD1 is interfering with RNA transcription and processing, including nuclear gemin functioning. The association of mutant hSOD1 with chromatin might sensitize the activation of p53 [83]. It is noteworthy that mutant hSOD1 in fruit flies causes chromosome aberrations [84]. More experiments, including chromatin immunoprecipitation and RNAseq and nuclear-specific deletion of SOD1, need to be conducted to explore these exciting new possibilities for MN degeneration in ALS.

## 4. Materials and Methods

### 4.1. Tg Mice

We used two tg mouse lines in this study. One mouse line (B6SJL-TgN[SOD1-G93A]1Gur/J, G1H line, stock #002726, The Jackson Laboratory, Bar Harbor, ME, USA) had a high copy number of hSOD1-G93A mutant allele (~20 copies) and a rapid disease onset [70]. The other tg mouse line expressed the normal human WT *SOD1* gene (B6SJL-TgN[SOD1] 2Gur, stock #002297, The Jackson Laboratory) at levels comparable to the mutant protein in hSOD1-G93A mice [70]. Non-tg age-matched littermates were used as controls for the indicated experiments. The number of individual mice per group generally was 4–7. The institutional Animal Care and Use Committee approved the animal protocols.

### 4.2. Immunohistochemistry on Mouse Spinal Cord

Mice were euthanized by carbon dioxide gas inhalation and were fixed in situ by intracardial perfusion of 4% paraformaldehyde for 20 min. The tissues were then removed and postfixed overnight and then cryoprotected in 20% glycerol for 48 h at 4 °C. Lumbar spinal cord sections were cut at 40 µm using a sliding microtome. The sections were stored in an antifreeze buffer at −20 °C until used.

The sections were washed overnight in tris-buffered saline (TBS, 50 mM Tris: 150 mM NaCl) at 4 °C to remove the antifreeze buffer. They were then permeabilized in 0.4% Triton X-100 (Tx) in TBS for 30 min, blocked in 10% normal goat serum/0.1% Tx in TBS for 60 min, and incubated in primary antibody in blocking solution for 48 h at 4 °C. The primary antibodies used were as follows: goat polyclonal anti-ChAT (Millipore, Burlington, MA, USA, 1:100); mouse monoclonal anti-hSOD1 (MBL, Ottawa, IL, USA, 1:100); rabbit polyclonal anti-parvalbumin (Abcam, Cambridge, UK, 1:500); rabbit anti-glutamine synthetase (LSBio, Seattle, WA, USA, 1:500); and rabbit anti-2′,3′-cyclic nucleotide 3′-phosphodiesterase (CNPase, LSBio, 1:500) as a marker of the oligodendrocytes. These primary antibodies have been characterized by us [22,52,53,85,86]. The sections were then washed in TBS and then incubated in species-appropriate secondary antibody (1:400) conjugated to Alexa-Fluor-488, -594, or -647 (Molecular Probes, Eugene, OR, USA), washed in TBS containing Hoechst-33342 dye (Molecular Probes) for nuclear staining in some instances, washed again, mounted with Vectashield (Vector Laboratories, Burlingame, CA, USA), and coverslipped for epifluorescence or confocal microscopy. The stained sections were viewed on a Zeiss LSM 510 Meta inverted confocal microscope with the appropriate lasers and filters, and the images were captured and analyzed using the LSM Zen 2009 image browser software (Zeiss, Jena, Germany). The negative control conditions were from non-tg mice and tg mice that were exposed to species-identical, IgG isotype-specific, non-immune IgG substituted for the primary antibody, with all other steps being identical to those used for the primary antibody. Image processing was conducted using the LSM Zen 2009 software with slight alterations of brightness and contrast without changing the content and actual results.

For immunoperoxidase immunohistochemistry, the spinal cord sections were rinsed to remove the antifreeze buffer and permeabilized in 0.4% Tx in TBS for 30 min, blocked in 4% normal goat serum/0.1% Tx in TBS for 60 min, and incubated in primary antibody in the blocking solution for 48 h at 4 °C as follows: mouse monoclonal anti-hSOD1 (MBL, 1:1000); rabbit monoclonal anti-hSOD1 (Epitomics, Burlingame, CA, USA, 1:1000); rabbit polyclonal anti-parvalbumin (Abcam, 1:2000); mouse monoclonal anti-glutamine synthetase, clone 6, (BD Transduction Laboratories, San Diego, CA, USA, 1:4000); and mouse anti-CNPase (Millipore Sigma, Burlington, MA, USA, 1:2000). The sections were then washed in TBS and then developed using the Vectastain Elite ABC kit (Vector Laboratories) according to manufacturer’s protocol with diaminobenzidine (DAB) as the chromogen. Sections were mounted, counterstained with cresyl violet, and coverslipped. The negative control sections were from non-tg mice or sections exposed to species-identical, IgG isotype-specific, non-immune IgG substituted for the primary antibody, used at an identical µg/mL concentration, with all other steps being identical to those used for the primary antibody. The stained sections were viewed on an Olympus microscope at the indicated magnification, and the digital images were captured with a Nikon camera and ACT-1 software (V2.63).

Profile cell counting was conducted on immunofluorescence and immunoperoxidase immunohistochemical sections. Generally, cells were counted in the spinal cord sections that were prepared in a systematic random sample design, with one in ten sections being stained for each marker. Only the lumbar spinal cord was used, and about ten stained, anatomically matched sections from each mouse were counted for each marker. The counting methods are described in more detail elsewhere [22].

### 4.3. Preparation of Spinal Cord Tissue for Isolation of MNs and Cell Nuclei

The spinal cords were isolated from non-tg, hSOD1-G93A, and hSOD1-WT mice after lethal exposure to inhaled carbon dioxide and then decapitation. Lumbar enlargements were used. After removing the pia, the spinal cord enlargements were dissected segment-by-segment under a surgical microscope, and then the segments were microdissected into columns of ventral horn gray matter without appreciable contamination from the dorsal horn and the surrounding white matter funiculi [54]. Tissue samples were collected and rinsed in a cell culture dish on ice containing dissection medium (1× Ca^2+^ and Mg^2+^ free Hanks balanced salt solution, GibcoBRL, Grand Island, NY, USA, supplemented with glucose and sucrose). These tissues were used to isolate MNs.

### 4.4. Preparation of MN-Enriched Cell Suspensions and MN Nuclei

Ventral horn samples were digested (20 min) with 0.25% trypsin-EDTA (Gibco) in a tissue culture incubator (5% CO_2_ and 95% air at 37 °C) and then triturated gently with a transfer pipette (see Appendix A). The tissue digests were transferred to a 5 mL centrifugation tube on ice, and the remaining small pieces of gray matter were further digested in trypsin-EDTA (~15 min). The total cell suspension was then subjected to differential low-speed centrifugation for cell sorting to isolate a spinal cord MN-enriched fraction. Tissue digests were centrifuged (Beckman GPR model centrifuge, GH 3.8 swinging bucket rotor) at 200 rpm (20 g_av_) for 5 min (4 °C) to remove white matter and blood vessels (P1 fraction), and then the supernatant (S1 fraction) was collected and centrifuged at 400 rpm (50 g_av_) to yield the S2 and P2 fractions. The supernatant (S2) was again collected and centrifuged at 800 rpm (160 g_av_) to yield the S3 and P3 fractions. After each spin (for 5 min), the pellet was resuspended in 100 µL phosphate-buffered saline (PBS, 100 mM phosphate:150 mM NaCl, pH 7.4) and fixed with 4% paraformaldehyde (4 °C for 1 h) for the immunophenotypic characterization of cell fractions. The primary antibodies used were used for immunofluorescence: mouse monoclonal anti-NeuN (Millipore, 1:200) and rabbit polyclonal anti-lamina-associated polypeptide-2 (Lap2, Santa Cruz, Dallas, TX, USA, 1:500) [49].

To isolate the nucleus-pure fractions of the spinal cord MN fractions, we used gentle trypsin digestion and differential centrifugation. MN pellets were washed in ice-cold Ca^2+^-free and Mg^2+^-free Hank’s balanced salt solution and then digested in 0.05% trypsin-EDTA at 37 °C for 5 min with gentle trituration. The volume of the cell digest was adjusted with Tris-NP40-EDTA and then centrifuged (Beckman GPR model centrifuge, GH 3.8 swinging bucket rotor) at 20 g for 5 min at room temperature. The P1 pellet, containing intact MNs, undigested cell aggregates, and residual tissue debris, was discarded, and the supernatant (S1) was collected and centrifuged at 400 rpm (50 g_av_) for 5 min at room temperature. The P2 pellet was saved and resuspended in Tris-NP40-EDTA, and the S2 supernatant was diluted in Tris-NP40-EDTA. These fractions were centrifuged at 1000 rpm (250 g_av_) at room temperature; the pellets (containing cell nuclei) were combined and resuspended in Tris-NP40-EDTA and centrifuged again at 1000 rpm to wash away trypsin-EDTA. The pellet was resuspended in Tris-NP40-EDTA and digested for 5 min at 37 °C with 2.5 mg/mL proteinase K-agarose (Sigma-Aldrich, St. Louis, MO, USA) and then centrifuged at 20 g_av_ to remove the proteinase K-beads. The supernatant containing cell nuclei (confirmed by microscopy) was collected and centrifuged at 14,000 g_av_ to collect the nuclei that were then resuspended in Tris-20% glycerol with protease inhibitors. The nucleus-enriched fractions were assayed for relative purity by immunoblotting, bright field microscopy, and immunofluorescence. For immunofluorescence, a small aliquot was allowed to dry on a gelatin-coated slide prior to staining.

### 4.5. Preparation of Brain and Spinal Cord Subcellular Fractions

Spinal cords from non-tg and hSOD1-G93A mice were used to assess the subcellular compartmentation of hSOD1 in vivo. The tissue subcellular fractionation was conducted by differential detergent extraction using a commercial kit (Thermo Fisher, Waltham, MA, USA, Subcellular Fractionation kit, 87790) as directed by the product protocol. We validated the tissue fractionation into the cytoplasmic extract (CE), soluble nuclear extract (sNE), and chromatin nuclear extract (cNE) using immunoblotting for the subcellular markers. This method was conducted specifically because of our interest in hSOD1 in subnuclear compartments of tg mice. This method was incompatible with other experiments designed to assess intact nuclei from mice.

### 4.6. Immunoblotting

Protein concentrations were measured by a bicinchoninic acid (BCA) protein assay with bovine serum albumin (BSA) as standard (Thermo Fisher, Pierce, BCA Protein Assay kit). Tissue and cell fractions were separated by SDS-PAGE on 4–12% NuPAGE gels (Invitrogen, Eugene, OR, USA) or on 16% non-precasted gels under denaturing and reducing conditions. The proteins were transferred to a nitrocellulose membrane using a Trans-Turbo system (Bio-Rad) according to the manufacturer’s protocol. The membranes were stained with Ponceau S (Sigma-Aldrich) to validate transfer efficiency and to image. The membranes were then destained with TBS and blocked with 1% BSA or 2.5% non-fat dry milk in TBS/0.05% Tween 20 for 1 h, and then incubated with primary antibody in the blocking solution overnight at 4 C. For the spinal cord and brain subcellular fractionation immunoblot experiments, the following antibodies used were: mouse monoclonal anti-hSOD1 (MBL, 1:1000); mouse monoclonal antibody to GAPDH (RDI, Waco, TX, USA, 1:10,000); mouse monoclonal antibody to methyl-CpG-binding protein 2 (MeCP2, Novus Biologicals, Centennial, CO, USA, 1:1000); and mouse monoclonal antibody to histone3 H3 (Cell Signaling Technologies, Danvers, MA, USA, 1:1000). For the nuclear fraction Western blots, the following antibodies used were: mouse monoclonal antibody to nitrotyrosine (Millipore, 1:1000); RAC1 (Cytoskeleton, Denver, CO, USA, 1:2000); and p47phox (Santa Cruz Biotechnology, 1:1000). After primary antibody incubation, the blots were washed in blocking buffer, incubated with species-appropriate HRP-conjugated secondary antibody (Bio-Rad, Hercules, CA, USA) diluted at 1:10,000–50,000 in the blocking solution, washed in blocking buffer, and were developed with ECL Supersignal West Pico reagent (Thermo Scientific, Rockford, IL, USA). Immunoreactivities were visualized on the membranes using a CCD camera and BioRad ChemiDoc Imaging System with Image Lab Touch Software 4.0 (version 3.0.1) or X-ray film. The Western blots were conducted in triplicate or greater.

### 4.7. Comet Assay

The comet assay was used to profile the DNA damage in MNs from non-tg and hSOD1 tg mice. The method details can be found elsewhere [54,55]. Cell microgels were prepared on slides. The procedures for the comet assay were conducted under low light to minimize environmentally induced DNA damage. The cell microgels were prepared as layers, as described, with low-melting point agarose and MN cell suspensions. After setting, the cell microgels were covered with lysis buffer (pH 10) containing 2.5 M NaCl, 100 mM EDTA, 1% sodium lauryl sarcosine, 10 mM Tris, and 1% Triton X-100. The cell microgels were lysed for 30 min (at room temperature). After draining, the microgels were treated with DNA-unwinding solution (300 mM NaOH, 1 mM EDTA, at pH 13) for 30 min (room temperature). The gels were run with constant current (300 mA at room temperature) for generally 20 min. After electrophoresis, the microgels were neutralized with 50 mM Tris-HCl (pH 7.5) for 15 min (twice). DNA was visualized with ethidium bromide staining. The comet evaluation and image acquisition were performed using a Zeiss Axiophot microscope. The comet profiles (Figure 9) were analyzed by counting the number of MN nuclei with comets. The number of comets and large intact cell nuclei regarded as MNs, stained by ethidium bromide, were counted in six microscopic views at 200× from the microgels. The percentages of comets relative to the total number of cells (total number of comets and total number of intact cell nuclei) were determined, and the group means were calculated.

### 4.8. NCF1 Gene Knockout Experiment

To determine if NADPH oxidase participates in the mechanisms of disease in hSOD1-G93A mice, we conducted a gene knockout study targeting the gene encoding p47phox. The hSOD1-G93A mice were crossed and backcrossed with *NCF1* gene-ablated mice (B6N.129S-Ncf1^tm1Shl^/J, The Jackson Laboratory, strain #027331) to generate hSOD1-G93A mice with the homozygous deletion of *NCF1*. Genotyping was conducted by standard PCR using the primers and protocol established by The Jackson laboratory. hSOD1-G93A littermates with both *NCF1* alleles were used as the controls. Lifespan was the outcome measurement. The group sizes were 12–15 mice.

### 4.9. Pharmacological Intervention Experiment with Apocynin

To use an experimental design more translationally relevant than gene targeting, we conducted a pharmacological study using apocynin (Sigma-Aldrich) as an inhibitor of NADPH oxidase [57,59,61,79] at a dosage of 10 mg/kg (ip) daily, starting at 7 weeks of age. The hSOD1-G93A mice receiving equal volumes of vehicle (saline/cyclodextrin/DMSO) were used as controls. The starting number of mice/group was 12. Equal numbers of males and females were used in each group. Lifespan was the outcome measurement.

### 4.10. Human Tissues

CNS tissues (Table 1) were obtained from the Human Brain Resource Center at JHMI. The institutional IRB and Health, Safety, and Environment Committee (JHU registration B1011021110) approved the use of postmortem human tissues. The protocol met all ethical and safety standards. De-identified postmortem samples of the brain (cerebral cortex Brodmann areas 4 and 3) and spinal cord were from patients with either sporadic ALS or familial ALS (Table 1). De-identified, aged, human control CNS tissues were from individuals without neurological disease (Table 1). The ALS patients were diagnosed by neurological examination using the El Escorial criteria [13]. The groups were matched for age and postmortem delay (Table 1). Cases were randomly obtained as autopsies occurred, and accessioning was independent of gender and race; therefore, males, females, and minorities were represented. Formalin-fixed, paraffin-processed tissue was used for the immunohistochemical studies of SOD1. The immunohistochemical negative controls were sections that were incubated with normal non-immune purified IgG at the same µg/mL concentration as that used for the primary antibody. DAB was the chromogen. Two sections of the lumbar spinal cord and motor cortex from each individual were counted, as described in detail [39].

### 4.11. Human iPS Cells and Directed Differentiation

The human iPS cell lines used in this study are shown in Table 2. CRISPR-Cas9 genome editing technology was used to introduce the SOD1-G93A missense mutation in a healthy control iPS cell line (C3-1), as described [51]. The genome editing of each cell clone was confirmed by DNA Sanger sequencing. Genetic off-target effects were also analyzed [51].

The directed differentiation of pluripotent stem cells (PSC) to mature MNs [51] was conducted along the course of intermediary cell types, including neuroepithelial cells (NEP) and MN progenitors (MNP), with each stage rigorously characterized by immunophenotyping using cell-specific antibodies.

### 4.12. Quantitative and Statistical Analyses

Mouse lumbar spinal cord tissue sections, stained by immunofluorescence and immunoperoxidase, were used for the profile counting of positive cells identified by cell-specific markers with hSOD1 nuclear positivity and nuclear ChAT or parvalbumin positivity. The ratios of nuclear hSOD1-positive/total cell type were determined. The mouse group sizes were 5–7 mice/genotype. The in vitro nuclear superoxide staining with hydroethidium was conducted on 4 mice/genotype/age. Comet assays were conducted on 4 mice/genotype/age. Western blotting for nitrotyrosine, p47phox, and RAC1 was conducted on 4–5 mice/genotype. Protein immunoreactivity was visualized with enhanced chemiluminescence, captured by direct digital imaging or X-ray film, and converted to digital images. Quantification of immunoreactivity was conducted using ImageJ 1.x, normalized to an empirically determined invariant protein. The mouse survival effects of *NCF1* gene deletion and apocynin treatment were determined from 12–15 mice/group. Profile cell counting of neuronal and glia was conducted (at 600× magnification) on human ALS and control autopsy brain sections that were immunoperoxidase stained for SOD1 with DAB as the chromogen as well as counterstaining with cresyl violet.

GraphPad Prism 9.5.1 was used to analyze the data. After passing the Shapiro–Wilks tests for normality, the data were compared for significant differences using one-way ANOVA and Mann–Whitney post hoc testing, and for the two group comparisons, a Student’s *t*-test was used. The significance was set at *p* < 0.05.

## 5. Conclusions

This study implicates nuclear SOD1 in cell death and the pathological biology of ALS. However, nuclear SOD1 is likely to be only one of several players in the highly complex pathobiology of ALS. Other nuclear instruments in the MN degenerative and cell death process seem to be NADPH oxidase and superoxide, which are potentially complicit in mislocalized SOD1’s facilitated formation of peroxynitrite, leading to DNA damage. Concurrent anomalies, including nuclear SOD1 interference with the cholinergic or other neurotransmitter phenotypes of MNs, could also be participatory, defining the selective vulnerability of lower MNs that have never been clearly identified in human ALS. This study also reveals the values, difficulties, and complexities of utilizing mammalian animal and cell model systems of ALS, all within the framework of human ALS, notably patient postmortem CNS, to discover disease-relevant commonalities for therapeutic targeting.

## Figures and Tables

**Figure 1 ijms-25-09106-f001:**
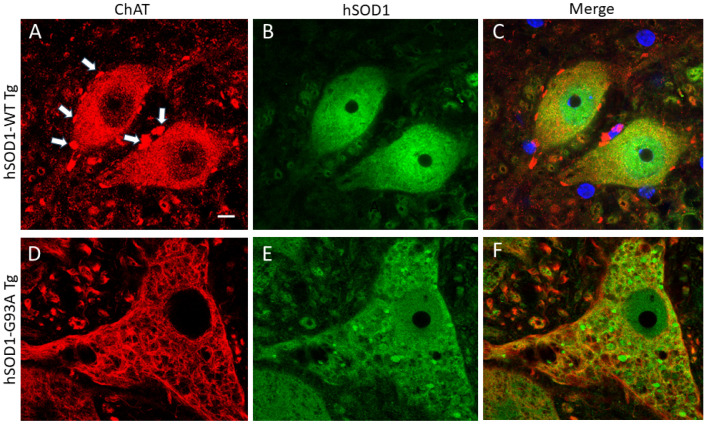
Immunofluorescence detection of ChAT and hSOD1 in tg mouse lumbar spinal cord. ChAT-positive (red) and hSOD1-positive (green) MNs are shown. The merged images importantly show both ChAT and hSOD1 immunoreactivities. (**A**–**C**) Spinal MNs in hSOD1-WT tg mouse. ChAT-positive C-boutons are identified (white arrows). Blue (**C**) is DAPI staining. Scale bar (in (**A**), applies to all panels) = 10 µm. (**D**–**F**) Spinal MNs in hSOD1-G93A tg mouse. The MN is severely vacuolated and swollen, as described previously [22], the ChAT nuclear staining is diminished, and C-boutons are lost. Non-tg mice are not shown because of the use of a human-specific SOD1 antibody. The images are representative of 5–7 mice per genotype.

**Figure 2 ijms-25-09106-f002:**
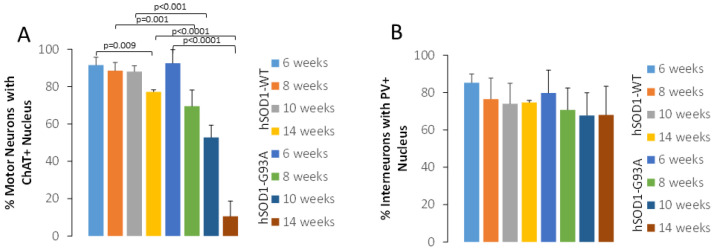
Quantification of MNs and interneurons in hSOD1-WT and hSOD1-G93A tg mice at different ages. Values are mean ± SD. Each mouse genotype at each age has 5–7 mice. (**A**) Graph showing the percentage of lumbar spinal cord hSOD1-positive MNs that have nuclear ChAT positivity in tg mice from 6 to 14 weeks of age. (**B**) Graph showing the percentage of lumbar spinal cord hSOD1/PV-positive interneurons that have nuclear PV positivity in tg mice from 6 to 14 weeks of age.

**Figure 3 ijms-25-09106-f003:**
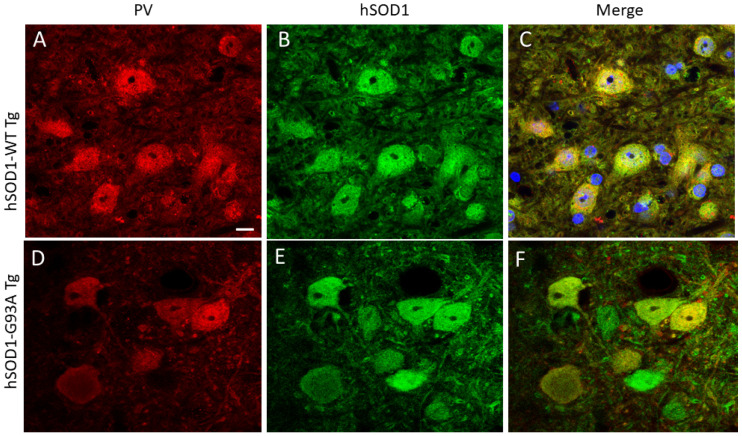
Immunofluorescence detection of the interneuron marker parvalbumin (PV, red) and hSOD1 (green) in tg mouse lumbar spinal cord ventral horn. PV- and hSOD1-positive cells are shown. The merged images show both PV and hSOD1 immunoreactivity. (**A**–**C**) Spinal ventral horn interneurons in hSOD1-WT tg mouse. Blue (**C**) is DAPI nuclear staining. Scale bar (in (**A**), applies to all panels) = 10 µm. (**D**–**F**) Spinal interneurons in hSOD1-G93A tg mouse. PV interneurons are depleted in hSOD1-G93A mice, as described [22], but the remaining PV neurons retain their nuclear PV positivity. Non-tg mice are not shown because of the use of a human-specific SOD1 antibody. The images are representative of 5–7 mice per genotype.

**Figure 4 ijms-25-09106-f004:**
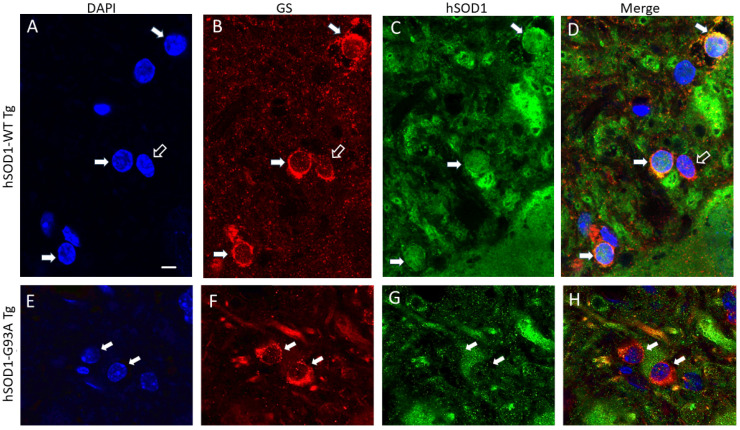
Immunofluorescence detection of the astrocyte marker glutamine synthetase (GS, red) and hSOD1 (green) in tg mouse lumbar spinal cord ventral horn. (**A**–**D**) Spinal ventral horn astrocytes in hSOD1-WT tg mouse. hSOD1^+^/GS^+^ astrocytes with a hSOD1^+^ nucleus (solid white arrow) and hSOD1^−^/GS^+^ astrocytes (open white arrow) are shown. Blue is DNA staining for nuclear discernment. Merged images show an overlap of DAPI, GS, and hSOD1. Scale bar (in (**A**), applies to all panels) = 10 µm. (**E**–**H**) Spinal ventral horn astrocytes in hSOD1-G93A tg mouse. SOD1^−^/GS^+^ astrocytes are shown (solid white arrows). Blue is DNA staining. The images are representative of 5–7 mice per genotype.

**Figure 5 ijms-25-09106-f005:**
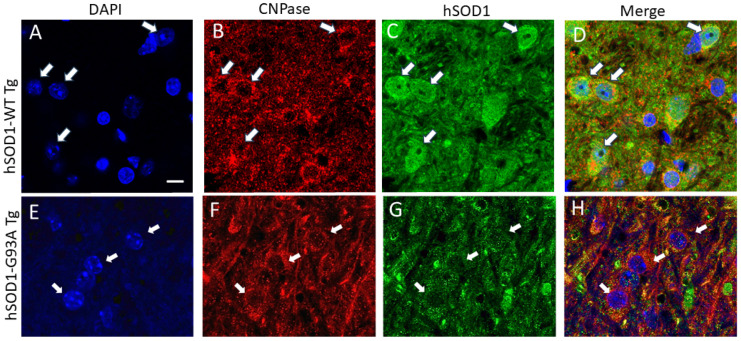
Immunofluorescence detection of the oligodendrocyte marker CNPase (red) and hSOD1 (green) in tg mouse lumbar spinal cord ventral horn. (**A**–**D**) Spinal ventral horn oligodendrocytes in hSOD1-WT tg mouse. hSOD1^+^/CNPase^+^ oligodendrocytes with a hSOD1^+^ nucleus (solid white arrow) are shown. Blue is DNA staining for nuclear visualization. Merged images show an overlap of DAPI, CNPase, and hSOD1. Scale bar (in (**A**), applies to all panels) = 10 µm. (**E**–**H**) Spinal ventral horn oligodendrocytes in hSOD1-G93A tg mouse. Oligodendrocytes lose hSOD1 positivity (solid white arrows). The images are representative of 5–7 mice per genotype.

**Figure 6 ijms-25-09106-f006:**
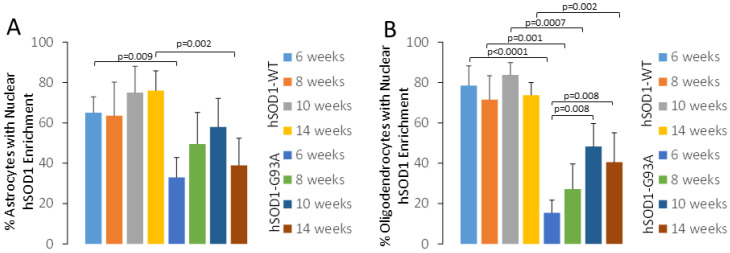
Quantification of astrocytes and oligodendrocytes in hSOD1-WT and hSOD1-G93A tg mice at different ages. Values are mean ± SD. Each mouse genotype at each age has 5–7 mice. (**A**) Graph showing the percentage of lumbar spinal cord ventral horn hSOD1-positive astrocytes that have nuclear hSOD1 enrichment in tg mice from 6 to 14 weeks of age. (**B**) Graph showing the percentage of lumbar spinal cord ventral horn oligodendrocytes that have nuclear hSOD1 enrichment in tg mice from 6 to 14 weeks of age.

**Figure 7 ijms-25-09106-f007:**
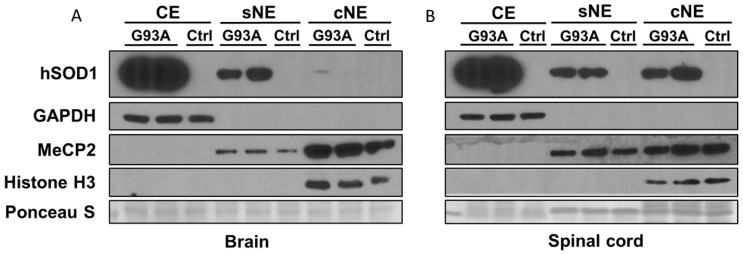
Western blotting of subcellular fractions of brains and spinal cords of hSOD1-G93A tg mice and non-tg mice. Subcellular fractions of cytoplasmic extract (CE), soluble nuclear extract (sNE), and chromatin nuclear extract (cNE) were characterized by subcellular compartment-specific protein enrichment: CE, glyceraldehyde phosphate dehydrogenase (GAPDH); sNE and cNE, methyl-CpG-binding protein 2 (MeCP2); cNE, histone H3. Ponceau S shows protein loading. (**A**) In the brains of hSOD1-G93A tg mice, hSOD1 was most enriched in the CE. Non-tg control mice had no hSOD1 immunoreactivity in any fraction. hSOD1 protein was detected in the sNE. In the cNE, only a faint trace of hSOD1 was detected. (**B**) In the spinal cords of hSOD1-G93A tg mice, hSOD1 was most enriched in the CE. Non-tg control mice had no hSOD1 immunoreactivity in any fraction. hSOD1 protein was detected in the sNE and cNE. Blots are representative of 4 mice/genotypes.

**Figure 8 ijms-25-09106-f008:**
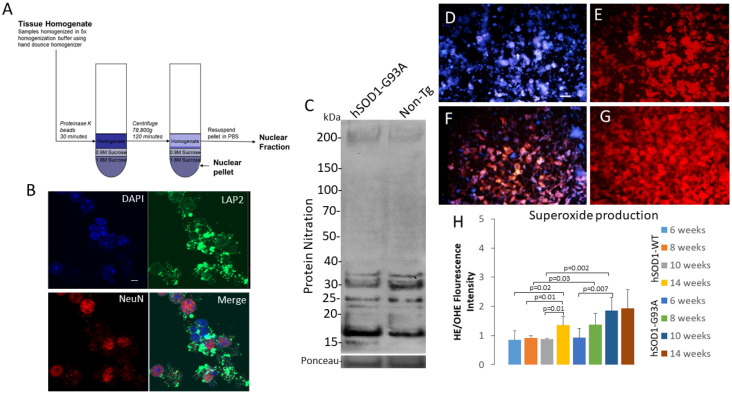
Spinal cord MN-enriched nuclear fractionation and detection of nuclear oxidative stress. (**A**) Diagram of protocol flow used to isolate and intact MN-enriched nuclear fraction for mouse lumbar spinal cord ventral horn using proteinase K digestion and sucrose gradient centrifugation. (**B**) Immunophenotyping of the nuclear fraction. DAPI staining for DNA shows large nuclei (scale bar in DAPI panel = 10 µm, applies to other panels) encircled by LAP2-positive (green) nuclear membrane and a nuclear matrix enriched in NeuN (red). (**C**) Western blot for nuclear protein tyrosine nitration in lumbar spinal cord ventral horn of hSOD1-G93A and non-tg mice. The mouse genotypes have specific protein nitration signatures with differential enrichment at low and high molecular weight ranges. (**D**–**G**) Detection of endogenous superoxide in living MN nuclei from hSOD1-WT (**D**,**E**) and hSOD1-G93A (**F**,**G**) mice. In the panel pairs of the exact fields, blue (**D**,**F**) shows unconverted hydroethidium, and red (**E**,**G**) shows superoxide-converted dye. Scale bar (in (**D**), applies to (**E**–**G**) = 100 µm. (**H**) Graph showing the superoxide production (indirectly indicated by the ratio of hydroethidium/oxidized hydroethidium) in MN nuclei isolated from hSOD1-WT and hSOD1-G93A tg mice. Values are mean ± SD (n = 4 mice/group).

**Figure 9 ijms-25-09106-f009:**
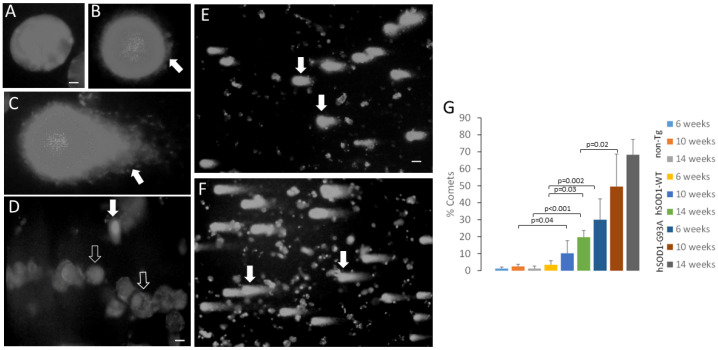
Comet assay demonstration of DNA single-strand break accumulation in spinal MNs of hSOD1 tg mice. (**A**–**F**) show ethidium bromide-stained nucleoids. (**A**) Nucleus with no DNA single-strand break accumulation. Scale bar (in (**A**), applies to (**B**,**C**)) = 1 µm. (**B**) Nucleus with emergent DNA single-strand break accumulation (DNA strands eluting at right side of nucleoid, white arrow). (**C**) Nucleus with moderate DNA single-strand break accumulation (white arrow). (**D**) Non-tg mouse (14 weeks old) showing one comet (solid white arrow). Many large MN nuclei with no DNA damage are present (open arrows). Bar (in (**D**), applies to (**E**,**F**)) = 10 µm. (**E**,**F**) Nuclear comet accumulation in hSOD1-WT and hSOD1-G93A tg mice spinal cord ventral horn MNs. (**G**) Graph showing the % comet accumulation in non-tg mice and hSDO1 tg mice at 6, 10, and 14 weeks of age. Values are mean ± SD (n = 4–5 mice/group).

**Figure 10 ijms-25-09106-f010:**
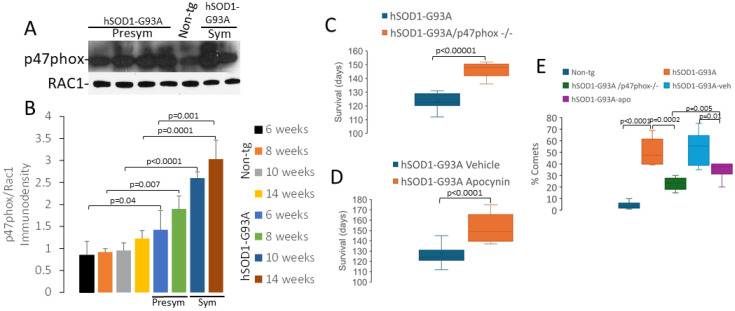
NADPH oxidase participates in the pathobiology of ALS in hSOD1-G93A mice. (**A**) Western blots of p47phox and RAC1 in spinal cord ventral horn of hSOD1-G93A and control mice. (**B**) Graph showing quantification of p47phox proteins levels in non-tg mice and in hSDO1-G93A mice at different stages of the disease. Values are mean ± SD (n = 4–5 mice/group). (**C**) Knockout of the *NCF1* gene encoding p47phox in hSOD1-G93A mice significantly extends lifespan. Box-and-whisker plots show mean survival with IQR and 5–95 percentile whiskers. (**D**) Apocynin treatment (10 mg/kg, ip) in hSOD1-G93A mice significantly extends lifespan. Box-and-whisker plots show mean survival with IQR and 5–95 percentile whiskers. (**E**) Comet assay DNA damage profiling shows that *NCF1* gene knockout and apocynin treatment hSOD1-G93A mice significantly attenuate DNA single-strand break formation in 10-week-old mice.

**Figure 11 ijms-25-09106-f011:**
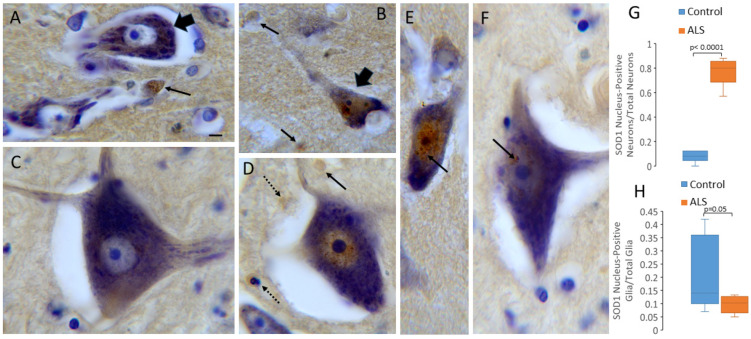
SOD1 localization in human ALS and age-matched control postmortem brains and spinal cords. SOD1 immunoreactivity was detected using the immunoperoxidase method with DAB as chromogen (brown) with cresyl violet counterstain (blue). (**A**) Control motor cortex: layer 5 Betz cells (broad black arrow) had cytoplasmic SOD1 immunoreactivity but low nuclear SOD1 immunoreactivity. Glial cells also had nuclear SOD1 immunoreactivity (thin black arrow). Scale bar (in (**A**), applies to all panels) = 7 µm. (**B**,**E**) ALS motor cortex: layer 5 Betz cells (broad black arrow) were attritional and had prominent nuclear SOD1 immunoreactivity. Many layer 5 neurons in ALS cases had SOD1 nuclear inclusions ((**E**), arrow). Glial cells were sometimes positive for nuclear SOD1 immunoreactivity and often attritional. (**C**) Control spinal cord: MNs had cytoplasmic SOD1 immunoreactivity and low nuclear SOD1 immunoreactivity. (**D**,**F**) ALS spinal cord: MNs were smaller than control MNs and had SOD1 nuclear immunoreactivity, some of which formed nuclear inclusions ((**F**), arrow). Some glia were positive for SOD1 ((**D**), thin black arrow), and many of these cells appeared apoptotic ((**D**), thin hatched arrows). (**G**) In ALS motor cortex layer 5, the number of neurons with prominent SOD1 immunoreactivity increases. Box-and-whisker plots show the mean with IQR and 5–95 percentile whiskers. (**H**) The number of motor cortex layer 5 glia with nuclear SOD1 did not change significantly in ALS cases compared to age-matched controls. Counts were conducted on 15 controls and 25 ALS cases (see Table 1).

**Figure 12 ijms-25-09106-f012:**
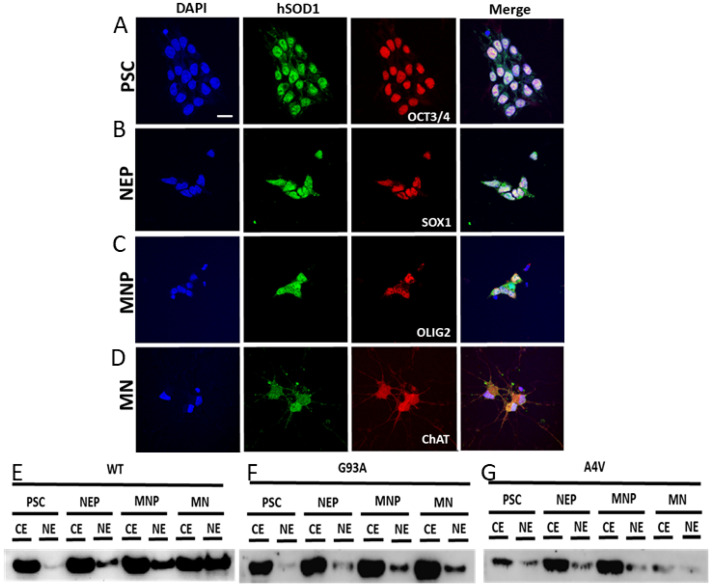
SOD1 localization and subcellular compartmentalization in human iPS cells, differentiated though stages to mature MNs (identified by ChAT) from pluripotent stem cell (PSC, identified by OCT3/4), neuroepithelial cell (NEP, identified by SOX1), and MN precursor (MNP, identified by Olig2). (**A**–**D**) DAPI shows the nucleus in relation to hSOD1 immunoreactivity (green) and different cell differentiation markers: OCT3/4, SOX1, OLIG2, and ChAT (red). All stages generally had nuclear SOD1 presence. (**E**–**G**) SOD1 presence in subcellular fractions of human iPS cells with SOD1-WT, -G93A, and -A4V alleles. Western blotting was conducted for SOD1 in cell cytoplasmic extracts (CE) and nuclear extracts (NE) at different stages throughout the course of directed differentiation to MNs. At the mature MN stage, the hSOD1-G93A and hSOD1-A4V genotypes were different from the hSOD1-WT genotype. Scale bar (in (**A**), applies to all panels) = 50 µm. Data are representative of three independent experiments. See Table 2 for cell lines used.

**Table 1 ijms-25-09106-t001:** Human autopsy cases used for brain and spinal cord samples.

Group	Case Number	Age (Years)/Sex	Postmortem Delay (Hours)	Cause of Death
Controls (Neurological Disease Free)	487	73/male	22	Pancreatic cancer
	515	62/male	21	Aortic aneurysm
	712	44/female	20	Pneumonia
	719	66/male	10	Myocardial infarction
	961	59/female	6	Myocardial infarction
	993	66/male	12	Prostatic carcinoma
	1277	80/female	8	Lymphoma
	1344	53/male	12	Metastatic carcinoma
	1348	44/male	18	Lymphoma
	1361	49/female	15	Thromboembolic disease
	1517	71/female	16	Heart disease
	1591	94/male	16	Pneumonia
	1603	89/male	16	Pulmonary embolism
	1613	74/male	4	Myocardial infarction
	1683	91/female	8	Cardiomyopathy
ALS	345	59/female	3	Respiratory arrest
	414	65/male	4	Respiratory arrest
	433	71/male	17	Respiratory arrest
	447	69/female	15	Respiratory arrest
	492	68/female	18	Respiratory arrest
	834F	46/male	3	Respiratory arrest
	875	70/female	24	Respiratory arrest
	950F	38/male	22	Respiratory arrest
	1014	72/male	5	Respiratory arrest
	1088	66/male	7	Respiratory arrest
	1108	64/female	8	Respiratory arrest
	1151	57/female	14	Respiratory arrest
	1161	47/male	6	Pneumonia
	1169	67/female	15	Respiratory arrest
	1176F	27/male	6	Respiratory arrest
	1359	61/female	14	Respiratory arrest
	1365	59/male	7	Respiratory arrest
	1386	69/female	15	Pneumonia
	1413	79/male	10	Respiratory arrest
	1452	65/female	6	Respiratory arrest
	1453	60/male	22	Pneumonia
	1485	61/female	5	Pneumonia
	1589	55/male	10	Pneumonia
	1614	69/male	18	Respiratory arrest
	1620	63/male	10	Respiratory arrest
	1623	71/female	59	Cardiopulmonary arrest
	1629	55/female	12	Pneumonia
	1635	68/female	5	Respiratory arrest
	1668	76/male	7	Respiratory arrest
	1693F	42/male	6	Respiratory arrest
	1713	54/female	14	Respiratory arrest
	1742	55/male	5	Pneumonia
	1755	72/female	5	Respiratory arrest
	1789F-SOD1A4V	55/male	14	Respiratory arrest

**Table 2 ijms-25-09106-t002:** Human-induced pluripotent stem cell lines were used.

iPSC Lines (Clone) ^1^	Gene	Mutation	Age of Donor	Gender
C3-1	Control	N/A	40	F
C3-1	SOD1	G93A	40	F
GO013	SOD1	A4V	63	F

^1^ These cell lines have been characterized [51].

## Data Availability

Data will be shared at the request of Lee J. Martin (martinl@jhmi.edu).

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
