# Peer review of "Nuclear Localization of Human SOD1 in Motor Neurons in Mouse Model and Patient Amyotrophic Lateral Sclerosis: Possible Links to Cholinergic Phenotype, NADPH Oxidase, Oxidative Stress, and DNA Damage"

_ijms, 2024, doi:10.3390/ijms25169106_

Round 1

Reviewer 1 Report

Comments and Suggestions for Authors

Martin and colleagues provided a very intersting and claryfing analysis of nuclear localization of hSOD1 in different ALS models and post-mortem human samples. 

Despite the impressive amount of work, this referee has some issues to raise and that should be addressed before publication.

1) I firmly believe that hSOD1WT-Tg animals could not be used as reference control. hSOD1WT may accumulate as well and determine toxic phenotypes that esulate from ALS. Authors should provide a non-tg animal as experimental control in order to dissect effectively the impact of hSOD1 mutation on the different analyzed phenotypes.

2) Figures 1 and 3 lack of DAPI staining. DAPI is only provided in the "Merge" panel that, per definition, should mix all the single channels that are showed, not others. Please add. Moreover, given that the work aims to demonstrate the NUCLEAR localization of hSOD1, DAPI staining is mandatory.

3) Please provide another astrocyte marker, e.g. GFAP.

4) Please provide another oligodendrocyte marker, e.g. MBP or RIP.

Minor concerns:

a) legends of figures 2 and 3 should be corrected, because there is no F letter in the caption.

b) FIgure 8 caption contains a mispelling error "hSDO".

Author Response

Reviewer 1.

Thank you for the astute comments.

1) “I firmly believe that hSOD1WT Tg animals could not be used as reference control.”

Response: We appreciate this view. We used hSOD1-WT transgenic (tg) mice as a comparator to the hSOD1-G93A tg mice for some of the experiments involving the localization of hSOD1 because non-tg mice do not have hSOD1. We used a human SOD1-specific antibody to detect the wt and mutant forms in the nucleus of specific types of cells in tg mice. We wished to see if WT SOD1 and mutant SOD1 behaved similarly at similar expression levels. Non-tg mouse sections are completely blank. For experiments not involving strictly the location of hSDO1, non-tg mice were used as controls.  Please see figures 7, 9, and 10 where non-tg animals were used as control comparators.

2. Figures 1 and 3 lack DAPI staining. DAPI is only provided in the ‘Merge’ panel…”

Response: We did not provide separate DAPI image panels in Figure 1 and 3 because the imaging for the microscopic fields that are shown were not optimal for in the DAPI channel and the nuclear DAPI of labeling of diseased motor neurons, like that shown in Figure 1F, is generally light with poor DAPI binding. We have described this before (Martin et al., 2007; Gertz et al 2012). In the figure legends of the revised manuscript, we clarify our meaning of the “merged” image.

3) “Please provide another astrocyte marker, e.g GFAP.”

Response: We used glutamine synthease (GS) as the astrocyte marker because it is an excellent astrocyte cell body marker without the confounding excessive neuropil staining that GFAP yields particularly in ALS mice which have profound GFAP-immunoreactive processes that can badly obscure cell bodies for counting. We wanted to draw attention clearly to the nuclear staining of the astrocytes that is why GS was selected as a marker. We have shown before that GS is an excellent marker for astrocytes (Tanagami et al., 2005). In the revised manuscript, we explain this rationale for using GS rather than GFAP.

4) “Please provide another oligodendrocyte marker, MBP or RIP.”

Response:  We used CNPase as the oligodendrocyte marker because it is an excellent oligodendrocyte cell body marker for the purpose of counting. In our experience MBP antibodies do not clearly label oligodendrocyte cell bodies in the mature CNS. The RIP antibodies that we have used have not withstood good validation as monospecific by western blotting in our hands. We have shown before that CNPase is an excellent marker for oligodendrocyte cell bodies that are particularly amenable to counting and that the antibody that we used is highly monospecific (Lee et al., 2021). In the revised manuscript, we explain this rationale for using CNPase rather than MBP.

Minor concerns a & b.

Response: we have corrected these errors.

Thank you for helping to improve the manuscript.

Reviewer 2 Report

Comments and Suggestions for Authors

The manuscript offers a detailed and rigorous exploration of the nuclear localization of human SOD1 in motor neurons in both mouse models and ALS patients. It presents compelling evidence linking nuclear SOD1 to oxidative stress and DNA damage, contributing to the pathogenesis of amyotrophic lateral sclerosis (ALS). The study's use of multiple experimental models and its translational relevance to human ALS provide a strong foundation. However, major revisions are necessary to enhance the clarity and depth of the findings.

1.     What is the sample size for each experimental age group?

2.     In the statistical analysis, the authors mentioned cell counting of neuronal and glia, but there was no information provided in the methods part.

3.     There are no n numbers and scale bars in all figure legends.

4.      For section 2.6, the results indicate that SOD1 localization in the nucleus of both motor neurons (MNs) and glial cells increases in ALS cases compared to controls. Can the authors elaborate on the pathological significance of this increased nuclear SOD1 presence in both cell types?

5.     For section 2.7, The study shows that while nuclear SOD1 is present in iPS cells at various stages of differentiation to motor neurons (MNs), familial mutant SOD1 ALS MNs exhibit aberrant subcellular compartmentation of SOD1. Can the authors provide more insights into the functional consequences of this aberrant compartmentation? How might these differences in SOD1 localization affect the cellular physiology and viability of MNs?

6.     The discussion highlights the novel findings regarding nuclear SOD1 localization and its potential role in ALS pathogenesis. What are the next steps in this line of research to further elucidate the mechanisms by which nuclear SOD1 contributes to motor neuron degeneration?

Comments on the Quality of English Language

None

Author Response

Reviewer 2

Thank you for your poignant and constructive comments.

  1. “What is the sample size for each experimental age group?”

Response: the sizes (n) of experimental group is given in the Materials and methods (page 15, 1st paragraph) and in each figure legend.

  1. “In the statistical analysis, the authors mentioned cell counting of neuronal and glia, but there was no information provided in the methods part.”

Response: In this revised manuscript this information is now provided in the materials and methods section (page 16, 1st paragraph and page 18, section 4.10).

  1. “There are no n numbers and scale bars in all the figures.”

Response: In the revised manuscript the group sizes (N’s) are stated in the figure legends. The scale bar allotment is now clarified. The scale bars sometimes shown in only one panel of a figures can apply to the other panels. This is now clearly stated in the figure legends of the revised manuscript.

  1. “For section 2.6,…can the authors elaborate on the pathological significance of this increased nuclearSOD1 in both cell types.”

Response:  In the revised manuscript in the discussion (section 3.5, page 14, last paragraph), we discuss this pathological significance.

  1. “For section 2.7, …Can the authors provide more insights into the functional consequences of this aberrant compartmentation.

Response: In the revised manuscript in the discussion (section 3.5, page 14, last paragraph), we also discuss the possible functional consequences of the aberrant SOD1 compartmentation.

  1. “…what are the next steps in this line of research….”

Response: In the revised manuscript in the discussion, for the sake of brevity, we also address this question in section 3.5, page 14, last paragraph.

Reviewer 3 Report

Comments and Suggestions for Authors

In the present manuscript, the Authors would to investigate the nuclear localization of SOD in different study models,  considering transgenic mice carrying wild type hSOD and hSOD G93A, human samples, and iPSC-derived motor neurons.

The Introduction well summarized the previous knowledge about this topic e well define the aims of the study.

Results are clear and well written. To ameliorate the comprehension of obtained findings, it would be better to put first the description of the result followed by the corresponding figure.

The Discussion analyzed obtained data according previous findings, evidencing limitation of the study and posing future research perspectives.

The "Material & Methods" section well explained the study design and the adopted methodologies. To let to the readers the reproducibility of the experiments, the Authors would indicate:

1. the model of the used Beckman rotor;

2. the speed in terms of x g (instead in rpm);

3. adopted dilution for all antibodies;  

4. the composition of all used  buffers;

5. the name of used commercial kits.

Indeed, to let a better comprehension of the subcellular fractionation, a schematic figure could be useful for this purpose: it could be reported in Supplemental Material. 

However, the adopted statistical approach is poor and seems to be not enough to explain the observed significance, because the Authors considered only the T-Student test. In this case, the use of one-way ANOVA with specific post-hoc tests would be more indicated, especially for the fact that two groups with different characteristics were considered at different stages. So, the Authors need to improve this section with an appropriate statistical analysis.

As a whole, the manuscript is well written and the topic is very interesting.

Author Response

Reviewer 3

Thank you for your astute and constructive comments.

  1. “the model of the used Beckman rotor:”

Response: In the revised manuscript, we state the details of the Beckman centrifuge (Section 4.4, page 16).

  1. “the speed in terms of x g (instead in rpm);”

Response: We have added g values for the centrifugations in the revised manuscript.

  1. “adopted dilution for all antibodies;”

Response: The revised manuscript has working dilutions for antibodies.

  1. “the composition of all used buffers;”

Response: We have added the buffer compositions in the revised manuscript.

  1. “the name of commercial kits," Response: we have added the names to all of the commercial kits used.

“A schematic figure would be useful for the subcellular fractionation”

Response: We have made a schematic figure for the cellular fractionation. It is included as Supplemental Figure 1.

“In this case, the use of one-way ANOVA with specific post-hoc test would be more indicated.

Response: We have redone the statistical analyses for the data sets using ANOVA and Mann-Whitney post-hoc testing. Some of the p values have changed and Figures 2, 6, 8, 9, and 10 have been revised.        

Round 2

Reviewer 1 Report

Comments and Suggestions for Authors

Please highlight your changes because this referee is not able to detect them in the manuscript. Moreover the suggested references were not added.

Author Response

Dear Reviewer,

Thank you for taking the time to reassess our revised manuscript.

We appreciate the additional reference suggestions. We have cited them in the revised manuscript.

The attached document pdf has the changes highlighted for your review. 

Thank you again for your time.

Reviewer 2 Report

Comments and Suggestions for Authors

The authors have answered all questions and made appropriate corrections to the original text. The current manuscript is more detailed than the previous one.

Author Response

Thank you for taking the time to re-review the manuscript.